cognition

predictive processing, visual search, spatial context, temporal context, contextual cueing

**Author for correspondence:**
Floortje G. Bouwkamp
e-mail: f.bouwkamp@donders.ru.nl

# No exploitation of temporal sequence context during visual search

Floortje G. Bouwkamp, Floris P. de Lange and Eelke Spaak

Donders Institute for Brain, Cognition and Behaviour, Radboud University, Nijmegen, The Netherlands

(iD) FGB, 0000-0002-6636-6393; FPdL, 0000-0002-6730-1452;
EP, 0000-0002-2018-3364

The human visual system can rapidly extract regularities from our visual environment, generating predictive context. It has been shown that spatial predictive context can be used during visual search. We set out to see whether observers can additionally exploit temporal predictive context based on sequence order, using an extended version of a contextual cueing paradigm. Though we replicated the contextual cueing effect, repeating search scenes in a structured order versus a random order yielded no additional behavioural benefit. This was also true when we looked specifically at participants who revealed a sensitivity to spatial predictive context. We argue that spatial predictive context during visual search is more readily learned and subsequently exploited than temporal predictive context, potentially rendering the latter redundant. In conclusion, unlike spatial context, temporal context is not automatically extracted and used during visual search.

## 1. Introduction

From the moment we wake up, we are continuously confronted with visual input. Selective attention helps us to navigate this rich visual environment, by enabling us to process what is important for the task at hand, and to ignore the rest. For instance, a bike ride to work requires us to attend to the road, the signs and traffic, without letting our eyes wander to the bright blue sky. Humans are incredibly good at focusing attention to select objects from cluttered scenes. An important factor that contributes to this ability is that our visual system makes use of predictable structure in our environment [1,2]. Our world might be very complex, but it is also very stable: the road is always beneath us and signs are usually found at the roadside. Covariation between objects and their environments creates a predictive context that can be exploited by the visual system to guide selective attention more

efficiently. There is indeed ample evidence that contextual expectations can help us identify objects in cluttered scenes [1–4]. Our visual system appears to be geared to automatically and implicitly extract regularities from our experiences [5,6] and use these regularities, or *predictive context*, to process the world more efficiently [7].

However, we navigate through the world not only in space, but also in time. Many events in life have a certain predictable order. On the aforementioned route to work, we use *spatial* context to locate traffic lights faster, but we also know, for example, that this particular roundabout will follow that specific turn. In other words, in everyday life, we might additionally make use of *temporal* context.

One way to investigate *spatial* predictive context is by using the well-established contextual cueing effect [7,8]. A contextual cueing experiment consists of a visual search task during which scenes, typically one T-shaped target embedded among similar looking distractors, are repeated. Though unaware of these repetitions, people become markedly faster in finding the target in these repeated scenes. The consensus is that contextual cueing leads to a more efficient guidance of spatial selective attention through the search displays [8–11].

To investigate *temporal* predictive context based on sequences, single items are generally shown in a specific order, i.e. as a pair, triplets, or longer sequences. Due to this order, one item becomes predictive of the following item(s). These items can be syllables [12], shapes [6] or objects [13–15]. In these studies, ranging from infants to primates, temporal predictive context typically leads to improved visual processing. Temporally predicted targets are more familiar, and detected or categorized faster [6,15] than unpredicted targets. In the brain, temporally expected images are accompanied by neural signatures of improved visual processing such as a higher dynamic range [13,14] or a suppressed BOLD response [15] compared with unexpected images. Similarly to the spatial regularities in contextual cueing, this temporal regularity typically goes unnoticed by participants.

It is unclear whether observers exploit sequence-based temporal predictive context, in addition to spatial regularities, when locating an item in a cluttered scene. Therefore, we set out to see whether the contextual cueing effect extends to the temporal domain. Can observers learn from both spatially predictive and temporally predictive context in visual search?

To answer this question, we exposed participants to various types of predictive context within a visual search task. As in classical contextual cueing, we repeated displays many times throughout the experiment, and intermixed them with completely novel displays. Therefore, we anticipated a behavioural benefit for the repeated scenes, replicating the classic effect of spatial predictive context. As a novel manipulation, we presented repeated displays either in a random or in a predictable order. Therefore, if observers additionally learn from temporal predictive context, we would expect a search time benefit when search scenes are repeated in a consistent order, compared with when they are repeated in a random order.

To preview our main results: we find a strong benefit of repeating scenes, but no additional advantage of presenting them in a consistent order. We conclude that, in the case of visual search, spatial predictive context might be dominant, rendering temporal predictive context redundant. This finding increases our understanding of how the visual system exploits previous experience to process the complex world more efficiently.

# 2. Material and methods

## 2.1. Participants

We recruited 40 healthy participants with normal or corrected-to-normal vision. Two participants were excluded as they missed too many trials (proportion late trials 38.54% and 29.98%, both scores fall below 25th percentile minus 1.5 × interquartile range), resulting in a final sample of 38 participants (27 women, age 19–36 years, $M = 24.58$, s.d. = 3.97). This sample size was chosen to yield greater than 80% power to detect differences with a medium effect size ($d = 0.5$) with a two-sided paired *t*-test at an alpha level of 0.05 (requiring $n = 34$), plus margin. The experiments were approved by the local ethics committee (CMO Arnhem-Nijmegen, The Netherlands) under the general ethics approval (CMO 2014/288, v. 2.1) and were conducted in compliance with these guidelines. All participants gave written informed consent beforehand and were paid for their participation.

## 2.2. Procedure

All stimuli measured $1.2° × 1.2°$ in size and were presented as black on a grey background. Participants were instructed to search for a T-shaped target stimulus among nine L-shaped distractors. Distractor L

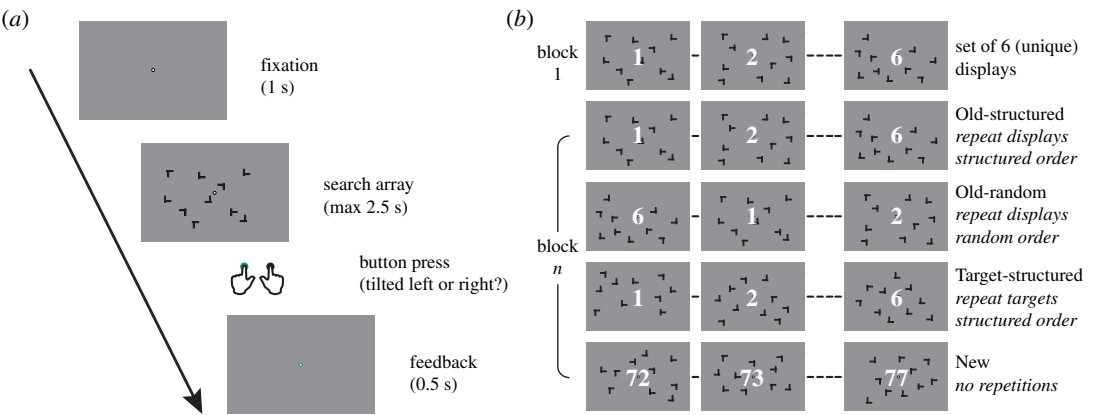

**Figure 1.** Search task and experimental design. (*a*) Schematic of an individual search trial. (*b*) The different conditions are defined by what is repeated on subsequent blocks (whole display or target location only) and in what order these are repeated (random or structured order). The same set of displays is shown here for illustration purposes only, in reality a unique set of displays was used for each condition.

shapes had a 10% offset in the line junction to increase search difficulty [16] and were rotated a random multiple of 90°. The target T was tilted either to the left or to the right (90°), and the participant was instructed to indicate the orientation of the T with a left or right button press. Throughout the experiment, a fixation dot (outer white diameter 8 pixels; inner black diameter 4 pixels) was presented at the centre of the screen. Participants were asked to fixate and not move their eyes. Each trial started with a 1 s fixation period, and search displays were presented until the response button press or up to 2.5 s, after which responses were too late. After the button press, participants were informed whether their response was on time and correct or not by the outer part of the fixation dot turning green (correct), red (incorrect) or blue (too late) for 0.5 s (figure 1*a*). A regular grid with steps of 1°, spanning from −9° to +9° horizontal and −6° to +6° vertical from the centre of the screen was used to arrange stimuli on the screen. Random jitter of ±0.4° was added to stimuli locations to prevent collinearities with other stimuli [8]. To ensure displays were of equal difficulty, the target always appeared between 6° and 8° of eccentricity, and the mean distance between the target and all distractors was kept between 9° and 11°. Additionally, both local crowding (too many distractors surrounding the target) and a 'standing out effect' (too few distractors surrounding the target) were not allowed. To further equate difficulty between the Old conditions (see Experimental conditions), we counterbalanced displays sets between the Old-random and Old-structured conditions across participants.

After practising the task for 72 trials, the main search task commenced, consisting of 36 blocks of 24 trials each. Participants were given feedback on their performance (% correct) and could take a short break every third block.

## 2.3. Experimental conditions

For each condition, six unique displays were repeated in a specific manner (figure 1*b*). Both the location and rotation of all distractors as well as the location of the target were the same throughout the experiment for *Old* displays. The tilt of the target (leftward or rightward) was, however, randomized to prevent participants from learning the correct motor response for a given display. This way, the layout of the distractors was only of predictive value for the target location, generating *spatial predictive context*. The six displays were represented together as a sub-block, but the order within the sub-block, however, was randomized for the Old-random condition. For Old-structured trials, this order was always the same. This predictable order added *temporal predictive context*. We also included a Target-structured condition. Instead of the entire configuration, only the location of the target was repeated across blocks for Target-structured displays, and this repetition of target locations was again in the same order across blocks. Target location on the first trial predicted the next target location, and so on, for all six Target-structured displays. All search blocks contained a sub-block of six 'New' trials, i.e. newly randomly generated search displays. Together this yielded four conditions, each containing six displays per sub-block. The order of these sub-blocks was randomly permuted within an experimental block, with the constraint that an immediate repetition of a condition was not allowed (i.e. the last condition of block *n* was not allowed to be the same as the first of block *n* + 1).

After the main search task, we probed explicit knowledge of the presented spatial and temporal regularities. Before testing participants' knowledge, we inquired about their subjective experience of recognition with the question: 'Did you have the feeling that some of the search displays occurred multiple times over the course of the experiment?' and indicated their (unspeeded) answer ('yes' or 'no') with a button press. Next, they were probed about their confidence: 'How sure are you about your answer to the previous question?' and answered with a button press ('very sure' or 'not very sure'). We pooled responses across confidence levels, since 74% (28/38) of participants indicated 'not very sure' to this confidence question. Subsequently, participants performed a recognition task. The 12 Old trials from the main experiment and 12 newly generated New trials were randomly intermixed and participants had to indicate whether a display was used during the main search task or not. Participants could take their time answering the question (no time-outs) and did not receive feedback.

Finally, participants were probed on their explicit knowledge of the structured order conditions. We inquired about their subjective experience with the question 'Some of the displays that were repeated were also always presented in the same order. Did you notice this?' and asked to answer with a button press ('yes' or 'no'). Again they were asked to indicate their confidence in the previous answer. After this, participants performed another recognition task, this time on the order of two displays that were presented consecutively. We presented 20 trials of two displays each from the structured order sets (Old-structured and Target-structured) and presented them either in the correct or incorrect order. Participants were instructed to indicate if the presented order on a given trial was the same as during the main search task, or different. Only four participants (10%) reported having noted a structured order of repetitions, and only half of them were confident in their answer. We, therefore, ignored both the subjective recognition rating and the confidence rating in the analyses of the data from this recognition task.

## 2.4. Apparatus

Stimuli were presented on a BenQ XL2420T monitor, at a resolution of $1920 \times 1080$ pixels and a refresh rate of 120 Hz, using Matlab (The Mathworks, Inc., Natick, MA, USA) and custom-written scripts using the Psychophysics Toolbox. Participants were seated in a separate room with dimmed light, at a distance of 55 cm from the screen using a chinrest.

## 2.5. Data analysis

Data analysis and visualization were done using R [17]. Reaction time (RT) was our primary, and accuracy our secondary variable of interest. Statistical assessment of the RT data was done for the second half of the experiment, i.e. blocks 18–36 (selected *a priori*). Only RT data from correct trials were used. For the structured order conditions (Old-structured and Target-structured), RT data for the first trial of each sub-block was discarded, as no temporal predictions can be made for the first display of the set. Since RT distributions are typically heavily skewed, the RTs per trial were log-transformed by taking the log10 value prior to any statistical analysis to improve normality.

To prevent participants from forming a more general prediction of target location per quadrant of the screen [18] we balanced target location per condition as much as possible across the quadrants of the screen, and regressed out any residual effect of quadrant frequency (QF) from the data per participant. For visualization purposes only, we smoothed the timecourses of both RT and accuracy across neighbouring blocks (i.e. the data plotted at block $n$ represents the mean of blocks $n-1$, $n$ and $n + 1$; figure 2a). All plots were generated with the ggplot2 package for R.

We report raw RTs in the Results section, but all statistical tests were done on log-transformed and QF-corrected quantities. If the assumption of sphericity was violated, as indicated by a significant outcome of Mauchly's test, we report the corrected $p$-value and degrees of freedom. All Greenhouse–Geisser $\varepsilon$ values were above 0.75; we, therefore, reported the more liberal Huyn–Feldt corrected values [19]. Effect sizes were calculated for all significant effects using the effsize package for R. For $F$-tests, this is the generalized eta-squared ($\eta_G^2$) measure of effect size [20]. For $T$-tests the value of Cohen's $d$ is computed using the approach of Gibbons *et al.* [21] for paired samples, including a suggested correction of Borenstein [22].

In addition to frequentist statistics, we report the Bayes factor. We applied a standardized Bayesian $t$-test, using the BayesFactor package for R. The Bayes factor quantifies the relative evidence for the alternative compared with the null hypothesis. The outcome is a ratio that quantifies how much more likely the data are under one hypothesis, than under the other. In our case, values greater than 1 mean the data are more likely under the alternative hypothesis than under the null hypothesis, and

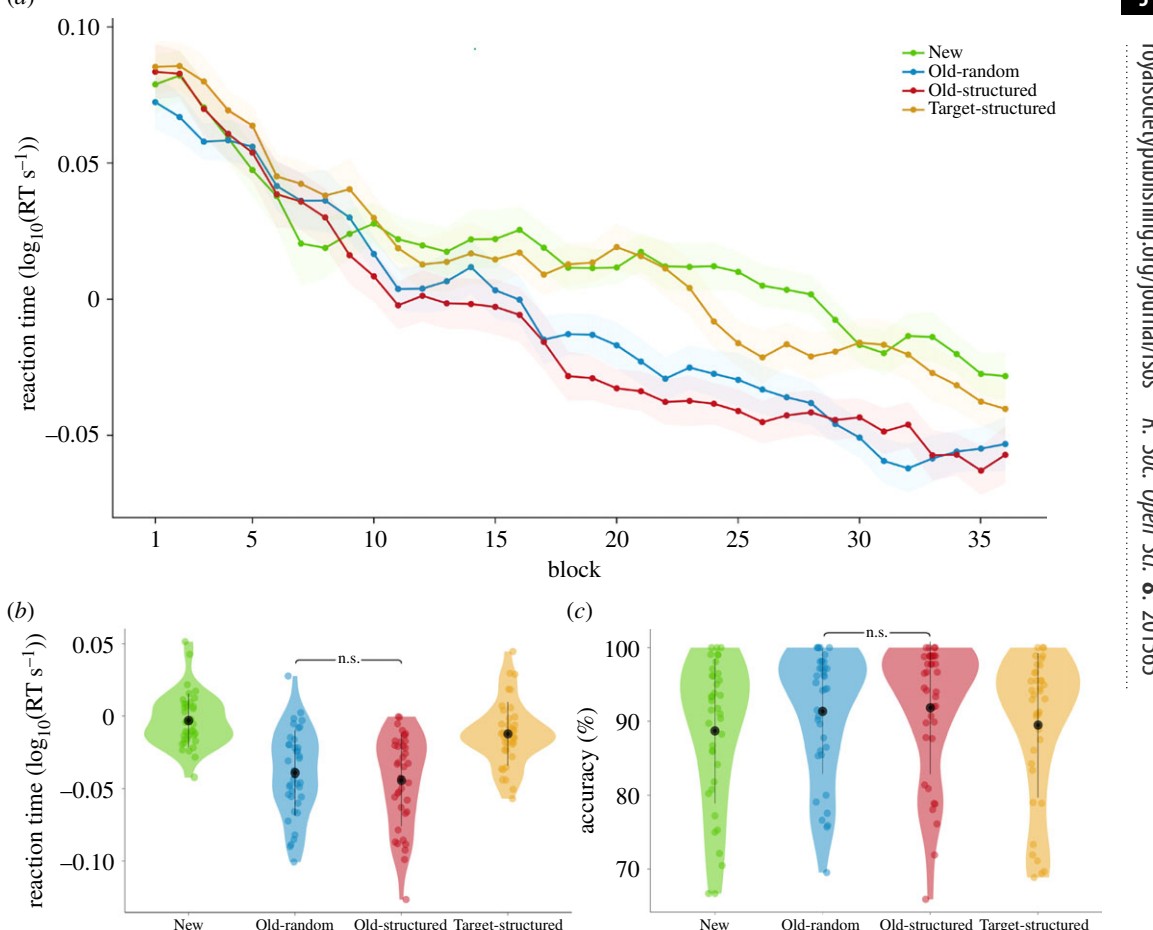

**Figure 2.** Results main search task. (*a*) Reaction time (smoothed across neighbours, by taking the mean of block $N$, $N-1$ and $N+1$) plotted over the timecourse of the experiment (shading indicates within-participant corrected standard error of the mean). (*b*) Reaction time and (*c*) accuracy within the second half of the experiment (block 19–36). Coloured dots are individual participants, the black dot reflects the mean and the black bar indicates $\pm 1$ standard deviation.

conversely, values less than 1 mean the data are more likely under the null hypothesis than under the alternative hypothesis.

# 3. Results

## 3.1. Visual search task

Both search speed and accuracy improved over the time course of the experiment (figure 2*a*), revealed by a main effect of time (first/second half, RT: $F_{1,37} = 174.4$, $p = 1.4 \times 10^{-15}$, $\eta_G^2 = 0.510$, accuracy: $F_{1,37} = 16.7$, $p = 2.3 \times 10^{-4}$, $\eta_G^2 = 0.027$). There was a significant difference between our conditions, both in RT and accuracy (RT: $F_{2.70,99.72} = 11.7$, $p_{HF} = 3.0 \times 10^{-6}$, $\eta_G^2 = 0.163$, accuracy: $F_{2.63,97.16} = 3.8$, $p_{HF} = 0.016$, $\eta_G^2 = 0.005$). Additionally, learning was revealed by an interaction between condition and time, again in both measures (RT: $F_{3,111} = 12.3$, $p = 5.3 \times 10^{-7}$, $\eta_G^2 = 0.048$, accuracy: $F_{3,111} = 4.1$, $p = 0.009$, $\eta_G^2 = 0.003$).

During the second half of the experiment, after learning has occurred, there was a clear benefit for the Old-random displays compared with the New displays (figure 2*b*,*c*). Participants were faster (Old-random: $1.16 \pm 0.44$ s, New: $1.28 \pm 0.46$ s, $t_{37} = 6.4$, $p = 1.9 \times 10^{-7}$, $d = 0.98$) and more accurate (Old-random: $91.34 \pm 8.51\%$, New: $88.70 \pm 9.80\%$, $t_{37} = -3.7$, $p = 7.0 \times 10^{-4}$, $d = -0.10$ BF$_{10} = 41.81$) in locating the target. Bayesian analysis of these results reveal very strong evidence for this benefit in accuracy (BF$_{10} = 41.81$) and extreme evidence in RT (BF$_{10} = 8.2 \times 10^4$). We thus unequivocally find the classical contextual cueing effect.

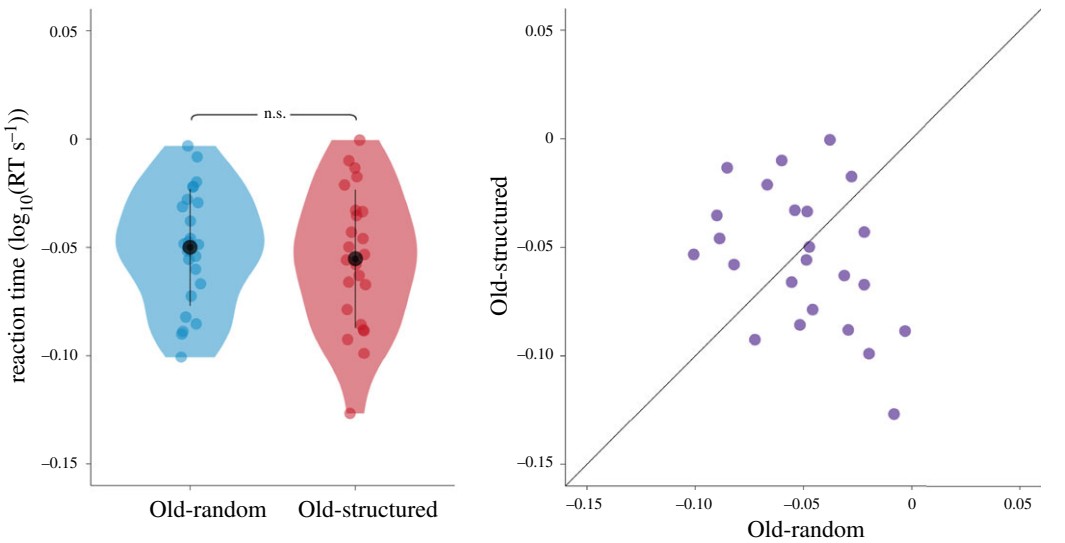

**Figure 3.** Individual differences in learning of Old scenes. Reaction time in the Old-random and Old-structured conditions of the second half of the experiment (block 19–36) for participants ($n = 24$) who were significantly faster on Old compared with New trials. Dots in all panels represent individual participants.

Additionally, when we compare the effect of a predictive target location (Target-structured) to no predictive context at all (New) we find no differences in accuracy (Target-structured: $89.50 \pm 9.80\%$, New: $88.70 \pm 9.80\%$, $t_{37} = -1.4$, $p = 0.17$). However, we do find a marginal effect in RT (Target-structured: $1.24 \pm 0.45$ ms, New: $1.28 \pm 0.46$ ms, $t_{37} = 2.0$, $p = 0.05$). A Bayesian analysis indicated inconclusive evidence for the presence of an effect on RT ($BF_{10} = 1.02$) and anecdotal evidence for the null hypothesis of no effect for accuracy ($BF_{10} = 0.43$). Considering we do find a strong contextual cueing effect, this indicates that the effect of spatial predictive context far exceeds any effect of temporal predictive context, when both operate in isolation. Participants were indeed markedly faster finding the target in scenes with spatial predictive context compared with when only target locations were predictive (Old-random: $1.16 \pm 0.44$ ms, Target-structured: $1.24 \pm 0.45$ ms, $t_{37} = -3.9$, $p = 3.6 \times 10^{-4}$, $d = -1.03$). The same was true for accuracy (Old-random: $91.34 \pm 8.51\%$, Target-structured: $89.50 \pm 9.80\%$, $t_{37} = 2.8$, $p = 8.0 \times 10^{-3}$, $d = 0.20$). A Bayesian analysis clearly favours the alternative hypothesis ($BF_{10} = 77.00$ for RTs and $BF_{10} = 5.00$ for accuracy).

Importantly, testing the key hypothesis of our experiment, we find no benefit of presenting scenes in a predictable order. Participants were on average equally fast in finding the target in both type of Old displays (Old-random: $1.16 \pm 0.44$ ms, Old-structured: $1.15 \pm 0.44$ ms, $t_{37} = 0.7$, $p = 0.50$). We find similar results for accuracy (Old-random: $91.34 \pm 8.51\%$, Old-structured: $91.83 \pm 9.01\%$, $t_{37} = -0.6$, $p = 0.56$). The Bayesian analysis of these results ($BF_{10} = 0.22$ for RT and $BF_{10} = 0.21$ for accuracy) indicate that the data are 4–5 times more likely under the null than under the alternative hypothesis. From this, we can conclude that there is no effect of adding temporal predictive context to spatial predictive context in our experiment.

It might be that temporal predictive context requires time to be recognized, and only impacts RT later in the sequence. To examine this possibility, we analysed the effect of temporal context in structured conditions per display (Old-structured per display versus Old-random overall; Target-structured per display versus New overall). Neither of these effects depended on display number (Old-structured $F_{5,185} = 0.30$, $p = 0.91$, and Target-structured $F_{5,185} = 1.07$, $p = 0.38$).

It is possible that the tendency to benefit from both temporal and spatial predictive context is governed by individual differences. Perhaps only people who show sensitivity to spatial predictive context are likely to be sensitive to temporal predictive context. To explore this possibility, in a follow-up analysis, we focused specifically on those participants sensitive to spatial predictive context (i.e. those with a contextual cueing effect in general, as defined by a RT benefit for Old displays (Old-random/Old-structured combined compared with New). Within this group, we tested if there was a difference between the Old conditions (Old-random versus Old-structured). However, there were again no differences in the speed with which the target was found between Old-random and Old-structured displays ($n = 24$, Old-random: $1.11 \pm 0.17$ ms, Old-structured: $1.11 \pm 0.18$ ms, $t_{23} = 0.52$, $p = 0.61$) (figure 3). The Bayesian analysis revealed moderate evidence for the null hypothesis of no difference ($BF_{10} = 0.24$).

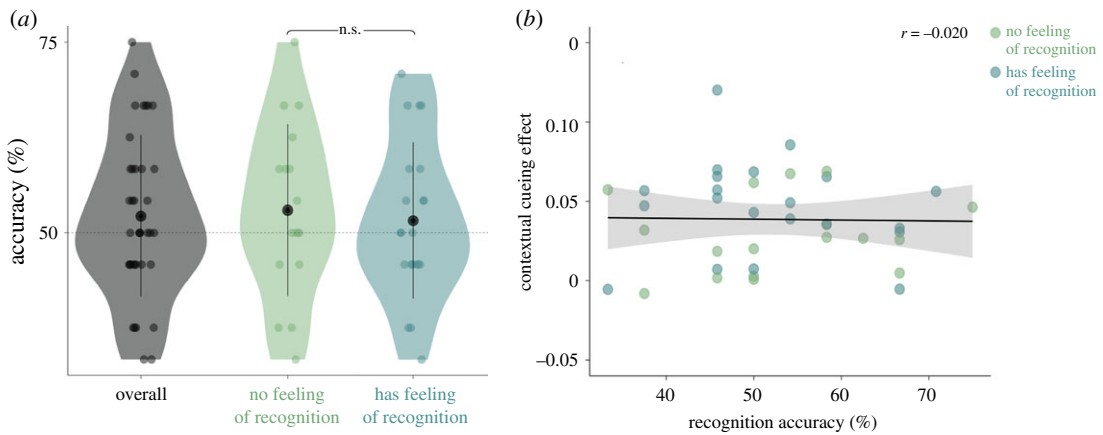

**Figure 4.** Results recognition task 1. (*a*) Accuracy in recognizing search displays as either Old or New for the whole sample (left, grey), and split by whether participants indicated a feeling of repeated displays during the main task (right, blue) or not (middle, green). (*b*) Contextual cueing effect during second half of the main task (New minus Old trials) for the same groups as in (*a*), as a function of recognition performance. Dots in both panels represent individual participants.

Taken together, these results demonstrate that people did not make use of temporal predictive context in addition to spatial predictive context during our visual search task. This lack of sensitivity to temporal predictive context is independent from individual sensitivity to spatial predictive context.

## 3.2. Recognition tasks

### 3.2.1. Task 1

Participants were at chance level in correctly identifying displays as Old or New (52.19 ± 10.60%, *t*-test versus chance level $t_{37} = 1.27$, $p = 0.21$, $BF_{10} = 0.37$). When probed before the recognition task, over half (21/38) of our participants reported to have a feeling that some displays were shown multiple times over the course of the experiment. However, these participants were not more accurate compared with participants who did not report such subjective feelings of recognition (feeling of recognition: 51.59 ± 10.25%, no feeling of recognition: 52.94 ± 11.29%, $t_{32.784} = 0.38$, $p = 0.70$, $BF_{10} = 0.34$). This is indicative of participants' subjective report of recognition not reflecting actual recognition performance. Neither of the two groups differed from chance (Feeling of recognition $t_{20} = 0.71$, $p = 0.49$, $BF_{10} = 0.29$, No feeling of recognition: 52.31%, $t_{16} = 1.07$, $p = 0.30$, $BF_{10} = 0.41$); we thus found no evidence of explicit memory for the Old displays. Additionally, we did not find a relationship between recognition accuracy and the contextual cueing effect (as defined by the difference between New and Old conditions, $r = -0.02$, $t_{36} = -0.12$, $p = 0.91$, $BF_{10} = 0.36$) (figure 4).

### 3.2.2. Task 2

Participants were at chance level for recognizing the order of two displays as correct or incorrect (52.89 ± 10.63%, *t*-test versus chance level $t_{37} = 1.68$, $p = 0.10$, $BF_{10} = 0.63$). A limited number of participants (4/38) reported to have a feeling that the order of some displays was repeated during the experiment. We, therefore, did not analyse these groups separately. Participants who were better at recognizing the order of displays, did not show a stronger effect of adding temporal to spatial predictive context: we did not find a relationship between recognition accuracy and the difference between random and structured order ($r = -0.0167$, $t_{36} = -0.10$, $p = 0.9208$, $BF_{10} = 0.36$) (figure 5).

## 4. Discussion

The aim of this study was to investigate the potential benefit of temporal predictive context over and above spatial predictive context during visual search. More specifically, we used a contextual cueing paradigm to investigate whether people can exploit temporal predictive context based on sequence order in addition to spatial predictive context. We found a clear and strong benefit for old displays

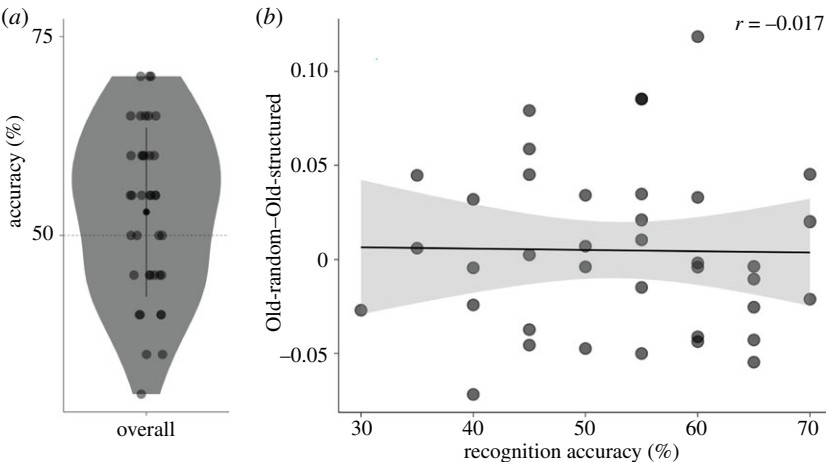

**Figure 5.** Results recognition task 2. (a) Accuracy in recognizing the order of two search displays as either correct or incorrect, for the whole sample. (b) The effect of adding temporal to spatial predictive context task (defined as RT difference between Old-random and Old-structured trials) during second half of the main task as a function of recognition performance. Dots in both panels represent individual participants.

compared with novel displays, confirming the strong influence of spatial predictive context during visual search. However, we did not observe an effect of repeating old scenes in a consistent order compared with presenting them in a random order. This was also true when we focused on only those participants who were sensitive to spatial predictive context. We thus found evidence that, using a typical visual search task, when locating a target in a complex scene, temporal predictive context is not exploited when the spatial predictive context is present. We discuss these findings below.

As anticipated, we were able to replicate the contextual cueing effect. This phenomenon is strong and robust and seen as evidence of spatial predictive context guiding selective attention through complex scenes [7,23]. Moreover, we found evidence neither for explicit knowledge of old scenes, nor for explicit knowledge of the order in which they were presented, as participants performed at chance level for all recognition tasks. This is thought to be evidence that contextual cueing is an entirely implicit effect [5,7,11]. Though this finding is in line with previous research, the support for the null hypothesis of chance-level performance, as expressed by a Bayes factor, was anecdotal. Interestingly, Spaak & de Lange [11] found that participants who revealed *more* explicit knowledge of repeated scenes had a *weaker* contextual cueing effect. We were, however, not able to replicate this negative relationship in the present study. An important difference between the two studies is that we had fewer old scenes: 12 instead of 20. Therefore, we might have been under-powered to detect above chance performance with this relatively small sample of displays [24]. In the Spaak and de Lange study participants who reported to have some subjective awareness of scenes being repeated, were better (and above chance) at recognizing old scenes. Although over half of our sample reported such a subjective feeling of awareness, these participants were not more accurate in recognizing old scenes, rendering this subjective report unreliable in our sample.

Our main finding is that sequence-based temporal predictive context was not exploited during our visual search task. It is important to acknowledge that visual search is an inherently spatial task. Even in the temporal predictive context conditions, spatial location(s) predicted spatial location(s) on the next trial. From this, it follows that in order to even detect temporal context, spatial context needs to be processed first. Therefore, perhaps only participants who are sensitive to spatial predictive context, are also sensitive to temporal predictive context. We explored this possibility by analysing participants who showed an effect of spatial predictive context separately. We, however, found no effect of temporal predictive context in this group of participants.

We found a marginal effect when only target locations were presented in a structured order (Target-structured). We do not feel confident interpreting this result, as it was only present in RT and not in accuracy, strictly speaking did not cross the threshold for significance ($p < 0.05$), and Bayesian analysis indicated inconclusive evidence. We included this condition to be able to address the question to what extent a possible effect of temporal predictive context (if found) elicited by the entire scene (Olds-structured) might be due solely to the repetition of the target, independent of the rest of the spatial configuration. By design, the Target-structured condition had a limited set of target locations

compared with the completely novel displays, which in turn would elicit a small amount of spatial predictive context. If the marginal benefit we observed indeed reflects a true effect, we believe this might explain it.

It might be that if the spatial context is fully predictive of the target location, temporal predictive context can no longer contribute in a significant way. This is in line with the notion of a stronger cue blocking the association with a weaker cue [25,26]. It would be interesting to see whether the temporal predictive context is taken into account when the spatial predictive context is more uncertain, for instance by changing the location of a subset of the distractors, which attenuates but does not abolish the contextual cueing effect [11,25,26]. This might additionally explain why our null finding is at odds with the phenomenon of intertrial contextual cueing [27,28]. This effect is a search benefit for finding a target on trial $N$, with the predictive context being an entire display (distractors and target location) on trial $N - 1$. Ono *et al.* [27] concluded that the 'visual system is sensitive to all kinds of statistical consistency and will make use of predictive information whether it is presented in a single trial or across trials'. In our case, however, the visual system seems to ignore the temporal consistency across trials. An important distinction to be made is that with intertrial contextual cueing, spatial predictive context is missing on trial $N$. The only context predicting target location on trial $N$ is, therefore, the spatial context on trial $N - 1$. If the spatial predictive context is dominant, this might explain why temporal predictive context *can* be exploited in intertrial contextual cueing: there is no spatial predictive context on trial $N$ 'hindering' the process. Given our results, we predict that adding spatial predictive context on trial $N$ in an intertrial contextual cueing experiment abolishes the effect. Supporting this notion, Olson & Chun [29] found that exposing participants to sequences of displays with multiple distractors, and a target at the end of the sequence, does yield a benefit in detecting the target when these sequences are structured compared with random. Future research is needed to establish this.

Finally, it could be that we did not use enough repetitions for participants to pick up temporal predictive context. We repeated the displays 36 times, which is plenty for the contextual cueing effect to appear, but we cannot exclude the possibility that more learning is required to pick up on temporal regularity. Research investigating temporal statistical regularities commonly uses equally many or even fewer repetitions [29–31], but an important difference might be that stimuli typically used are easily processed, such as objects in isolation. Visual search scenes are more complex, representing multiple target–distractor relations. Even though recent work shows there is also a component of scene memory to contextual cueing [32,33], this might not be strong enough to enable one 'scene' to function as a predictor of the next, analogous to how an object predicts the next one in typical temporal statistical learning tasks. A previous study exposed observers to sequenced information in addition to, but independent of, spatial predictive context. Signs of sequence learning were visible after 96 repetitions [34]. It is possible that this many repetitions of the spatial context in our study would have allowed exploitation of temporal predictive context. We question, however, whether learning will then remain implicit, introducing another source of variance. Nevertheless, it is a potentially interesting question: it could be that when temporal predictive context becomes explicit it will impact RTs via goal-directed attention [18]. On a very speculative note, temporal predictive context might even become an endogenous cue and block the more habitual effect of spatial predictive context [35].

Our results have implications for how we think about the role of predictive context in visual perception. Though both spatial and temporal context are of importance for visual perception [36], our findings imply that extracting and using context might not be as automatic as previously assumed. In our experiment, we find that spatial predictive context is favoured. This might be due to a modality-specific bias, leading to a dominance of spatial predictive context in visual perception [37–41]. However, sensitivity to one type of context over the other might also be task-dependent. Additionally, there are other types of temporal context, for instance, duration- or rhythm-based [29]. These temporal predictive contexts create expectations within the temporal domain itself: *when* the target will occur, instead of *where*. These types of temporal context might be exploited more efficiently in addition to spatial predictive context.

It is possible that temporal predictive context is encoded by observers, but not exploited. Within contextual cueing, it has been shown that the exploitation of spatial predictive context knowledge depends on attention, even though its acquisition does not [16,25]. Additionally, learned predictive context can be visible at the neural level without being expressed in behaviour [13,15]. Therefore, not finding an effect of temporal order in behaviour in our task does not necessarily imply that this predictive context is not encoded.

Concluding, in our study we found evidence that observers do not benefit from sequence-based temporal context on top of spatial predictive context during visual search. We argue that since people are focused on locating an object in space, spatial context is readily encoded and subsequently

exploited, while temporal context may not be. More research is needed to investigate the specific conditions that would allow for temporal contingencies to have an effect on search behaviour, and to ultimately understand how spatial and temporal predictability jointly shape our perception.

Ethics. The experiments were approved by the local ethics committee (CMO Arnhem-Nijmegen, The Netherlands) under the general ethics approval (CMO 2014/288, v. 2.1) and were conducted in compliance with these guidelines.

Data accessibility. All data and code used for stimulus presentation and analysis are released publicly under an open license at the Donders Repository: http://dx.doi.org/10.34973/hrcx-8w07 [42].

Authors' contributions. All authors designed research. F.G.B. performed research, analysed data, drafted the manuscript and made figures. All authors edited and approved the final version of the manuscript.

Competing interests. We declare we have no competing interests

Funding. This work was supported by The Netherlands Organisation for Scientific Research (NWO Research Talent grant no. 406.18.508 awarded to F.G.B., Veni grant no. 016.Veni.198.065 awarded to E.S. and Vidi grant no. 452-13-016 awarded to F.P.d.L.) and the EC Horizon 2020 Program (ERC starting grant no. 678286 awarded to F.P.d.L.).

Acknowledgements. We would like to thank Izana Bayerman for her help with collecting the data.

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
