## [Peer Review File · Royal Society Open Science]

Review History

RSOS-201565.R0 (Original submission)

Review form: Reviewer 1

Is the manuscript scientifically sound in its present form?

Yes

Are the interpretations and conclusions justified by the results?

Yes

Is the language acceptable?

Yes

Do you have any ethical concerns with this paper?

No

Have you any concerns about statistical analyses in this paper?

No

Recommendation?

Accept with minor revision (please list in comments)

Comments to the Author(s)

This study aims at investigating the effect of temporal regularities of events on human behavior. Using a modified version of the classic spatial contextual cueing paradigm, they show that temporal context – manipulated as the predictive order of search arrays – does not modulate response times. The experiment design and data analysis are carefully conducted, and the findings are unquestionably relevant to the field. I have only a data analysis suggestion and a more general comment.

A possible limitation of their analysis is that it's not possible to know when – within each sub-block – the temporal context would kick in. The authors somewhat acknowledge that, by excluding the first trial of each sub-block. However, I believe that there are better ways to deal with this potential problem. The simplest one would be to analyze RT performance separately for each display repetition (1 to 6). The other, perhaps more sensitive approach, would be to perform a regression analysis using the order as a predictor.

As a more general comment, the authors should be a bit more careful when they generalize their findings. Temporal context is an extremely broad concept, and I don't believe that a manipulation of the sequence order of events is enough to state that temporal context is not exploited, or extracted, during visual search. One of the original papers in this area (Olson and Chun, 2001), already noted the intricacies of the term: "Temporal structure can be divided into two time-based variables: (a) a sequential or ordinal variable and (b) a duration or rhythm-based variable." (pg. 1301). Therefore, I would strongly suggest to the authors to be more specific about the type of temporal prediction they investigated (sequence order) – including in the title and abstract.

Review form: Reviewer 2

Is the manuscript scientifically sound in its present form?

No

Are the interpretations and conclusions justified by the results?

No

Is the language acceptable?

Yes

Do you have any ethical concerns with this paper?

No

Have you any concerns about statistical analyses in this paper?

Yes

Recommendation?

Major revision is needed (please make suggestions in comments)

Comments to the Author(s)

Summary: This paper reports an experiment on the relationship between spatial and temporal predictive information in visual search. Participants did a T-among-L search task that had 4 conditions (across sub-blocks): (1) random order and configuration of all search items; (2) random order of 6 trials with repeated target and distractor locations; (3) repeated order of 6 trials with

repeated distractor and target locations; and (4) repeated order of 6 trials with repeated target locations (but not distractor locations).

This paper investigates an interesting question and is well-designed. I like this study, and I think the writing is strong: the introduction in particular is highly focused, which makes it easy to grasp the point of the study (with some minor exceptions I mention below), and the general discussion covers the relevant literature well. My concerns with this version of the study deal primarily with the analyses and some of the theoretical interpretation, both of which I think could be greatly improved with a revision to this submission.

Major Points:

1. I think your analysis of individual differences is inappropriate. By combining Old-ro and Old-so trials and comparing to New, then analyzing the difference within participants who show no Old vs. New difference, you are ensuring that on average participants in the “no contextual cueing” group show no Old-ro versus Old-so difference. The only way for them to show no Old vs New difference but a difference between Old-ro and Old-so is if one of these two Old conditions is worse than the new condition, something there is (to my knowledge) no theoretical reason to expect – this would only be due to noise. Therefore, I don’t think this test can appropriately assess whether participants who don’t show standard contextual cueing will show an advantage of repeating temporal context. Rather, you might use correlational analyses similar to those you show in Fig 3. For instance, you could compute the standard CC effect (Old-ro - New) and plot this against the Old-so effect (e.g., Old-ro - Old-so), which as far as I can tell avoids the concern I mention above. This is only one suggestion: there may be other, more appropriate ways to conduct these individual difference analyses. However, I do feel that a different approach to the one currently used is necessary.
2. Throughout the paper, I think you can be more precise regarding what your findings say about temporal predictability. The introduction seems to inconsistently discuss the effects of temporal predictability on visual search in general and the interaction between spatial and temporal predictability. I think revision of the last couple paragraphs of the introduction to lay out the fact that your experiment fully crosses manipulations of temporal predictability of targets (albeit target locations) with spatial predictability of entire search arrays. My impression is that you are addressing two interesting questions: (1) will temporal predictability help (in which case Targ-so condition should be faster than New and (2) will temporal predictability help when also in the presence of spatial predictability (Old-so faster than Old-ro). As currently written, your preview of the results makes it sound like you are only testing (2) and based on it concluding that temporal predictability is not useful in your task, while in reality you do present marginal evidence that temporal predictability is useful relative to no predictability, just that it is redundant with spatial predictability.

Minor points:

3. P4: the paragraph beginning “to investigate temporal” would benefit from a brief discussion of whether these past studies do or don’t find effects.
4. P5: I have no specific concerns about the encouragement that participants not move their eyes, but I do wonder why this was enforced in displays that were visible for a reasonably long time (2.5s), at least enough to make several saccades. Particularly because stimuli were not scaled with distance, this choice surprised me.
5. P6: I don’t believe the manuscript mentions the cutoff for “too late” trials, which might be good to include
6. P6: How do you think restricting target locations to between 6-8 degrees of eccentricity affects your results? Doing so is similar to making targets appear more often in one quadrant of a display (only for a ring of locations around fixation), something which has been shown to speed responses to targets within versus outside the ring (Druker & Anderson, 2010, *Frontiers in Human Neuro.*). This isn’t a confound – your results still report standard CC despite this change

from many CC studies – but I wonder if your results might change without that restriction in target locations (e.g., participants may be consciously guiding attention to the range of locations that could contain targets, making other implicit effects of attention less effective than if participants weren't engaging goal-driven attention; cf. Jiang, Swallow, & Rosenbaum, 2013, Guidance of spatial attention by incidental learning and endogenous cueing, JEP:HPP).

7. P7: It took me a long time to learn which condition names were associated with which conditions, so you might consider renaming them to something more intuitive (e.g., old-random, old-structured, target-structured, and new, or similar).
8. P8: Your log-transforming of all RT data together, rather than separately within each condition (and perhaps within each subject and condition), may be inappropriate. Could you explain your reasoning behind log-transforming all RT data independent of condition given the possibility that these data can be viewed as coming from different distributions assuming you do have an effect of condition?
9. P9: Fig 2a says the reported RT data is smoothed but not how this smoothing is performed. It would be good to state this.
10. P9: To be consistent, it would be good to report effect sizes for F tests as well as t-tests throughout.
11. P11: Related to my point about the discussion at the end of your introduction, I don't think your results indicate that "there is no effect of temporal predictive context in our experiment," but that "there is no effect of adding temporal predictive context to spatial predictive context in our experiment."
12. Do you think your results for the Targ-so condition would differ if the absolute serial order of target locations remained constant within each block (e.g., an experiment with the same sequence of target locations throughout an entire block)? I wonder if the need to identify when this sequence begins makes it very difficult to pick up on and/or guide attention based on this predictability.
13. P15: I recommend incorporating a discussion of Vadillo, Konstantinidis, and Shanks (2016, PBR) in your discussion of statistical power for explicit recognition tests, as it deals specifically with this question for contextual cueing tasks. This is only a suggestion.

Decision letter (RSOS-201565.R0)

Dear Ms Bouwkamp

The Editors assigned to your paper RSOS-201565 "No exploitation of temporal predictive context during visual search" have now received comments from reviewers and would like you to revise the paper in accordance with the reviewer comments and any comments from the Editors. Please note this decision does not guarantee eventual acceptance.

We do not generally allow multiple rounds of revision so we urge you to make every effort to fully address all of the comments at this stage. If deemed necessary by the Editors, your

manuscript will be sent back to one or more of the original reviewers for assessment. If the original reviewers are not available, we may invite new reviewers.

Please submit your revised manuscript and required files (see below) no later than 21 days from today's (ie 20-Nov-2020) date. Note: the ScholarOne system will 'lock' if submission of the revision is attempted 21 or more days after the deadline. If you do not think you will be able to meet this deadline please contact the editorial office immediately.

on behalf of Dr Isabelle Mareschal (Associate Editor) and Essi Viding (Subject Editor)
openscience@royalsociety.org

Associate Editor Comments to Author (Dr Isabelle Mareschal):

Associate Editor: 1

Comments to the Author:

Expert reviewers have raised some concerns about your paper, notably regarding the methodology / analysis. Please provide a point by point reply to their queries.

Reviewer comments to Author:

Reviewer: 1

Comments to the Author(s)

This study aims at investigating the effect of temporal regularities of events on human behavior. Using a modified version of the classic spatial contextual cueing paradigm, they show that temporal context – manipulated as the predictive order of search arrays – does not modulate response times. The experiment design and data analysis are carefully conducted, and the findings are unquestionably relevant to the field. I have only a data analysis suggestion and a more general comment.

A possible limitation of their analysis is that it's not possible to know when – within each sub-block – the temporal context would kick in. The authors somewhat acknowledge that, by excluding the first trial of each sub-block. However, I believe that there are better ways to deal with this potential problem. The simplest one would be to analyze RT performance separately for each display repetition (1 to 6). The other, perhaps more sensitive approach, would be to perform a regression analysis using the order as a predictor.

As a more general comment, the authors should be a bit more careful when they generalize their findings. Temporal context is an extremely broad concept, and I don't believe that a manipulation

of the sequence order of events is enough to state that temporal context is not exploited, or extracted, during visual search. One of the original papers in this area (Olson and Chun, 2001), already noted the intricacies of the term: “Temporal structure can be divided into two time-based variables: (a) a sequential or ordinal variable and (b) a duration or rhythm-based variable.” (pg. 1301). Therefore, I would strongly suggest to the authors to be more specific about the type of temporal prediction they investigated (sequence order) - including in the title and abstract.

Reviewer: 2

Comments to the Author(s)

Summary: This paper reports an experiment on the relationship between spatial and temporal predictive information in visual search. Participants did a T-among-L search task that had 4 conditions (across sub-blocks): (1) random order and configuration of all search items; (2) random order of 6 trials with repeated target and distractor locations; (3) repeated order of 6 trials with repeated distractor and target locations; and (4) repeated order of 6 trials with repeated target locations (but not distractor locations).

This paper investigates an interesting question and is well-designed. I like this study, and I think the writing is strong; the introduction in particular is highly focused, which makes it easy to grasp the point of the study (with some minor exceptions I mention below), and the general discussion covers the relevant literature well. My concerns with this version of the study deal primarily with the analyses and some of the theoretical interpretation, both of which I think could be greatly improved with a revision to this submission.

Major Points:

1. I think your analysis of individual differences is inappropriate. By combining Old-ro and Old-so trials and comparing to New, then analyzing the difference within participants who show no Old vs. New difference, you are ensuring that on average participants in the “no contextual cueing” group show no Old-ro versus Old-so difference. The only way for them to show no Old vs New difference but a difference between Old-ro and Old-so is if one of these two Old conditions is worse than the new condition, something there is (to my knowledge) no theoretical reason to expect – this would only be due to noise. Therefore, I don’t think this test can appropriately assess whether participants who don’t show standard contextual cueing will show an advantage of repeating temporal context. Rather, you might use correlational analyses similar to those you show in Fig 3. For instance, you could compute the standard CC effect (Old-ro - New) and plot this against the Old-so effect (e.g., Old-ro - Old-so), which as far as I can tell avoids the concern I mention above. This is only one suggestion: there may be other, more appropriate ways to conduct these individual difference analyses. However, I do feel that a different approach to the one currently used is necessary.
2. Throughout the paper, I think you can be more precise regarding what your findings say about temporal predictability. The introduction seems to inconsistently discuss the effects of temporal predictability on visual search in general and the interaction between spatial and temporal predictability. I think revision of the last couple paragraphs of the introduction to lay out the fact that your experiment fully crosses manipulations of temporal predictability of targets (albeit target locations) with spatial predictability of entire search arrays. My impression is that you are addressing two interesting questions: (1) will temporal predictability help (in which case Targ-so condition should be faster than New and (2) will temporal predictability help when also in the presence of spatial predictability (Old-so faster than Old-ro). As currently written, your preview of the results makes it sound like you are only testing (2) and based on it concluding that temporal predictability is not useful in your task, while in reality you do present marginal evidence that temporal predictability is useful relative to no predictability, just that it is redundant with spatial predictability.

Minor points:

3. P4: the paragraph beginning “to investigate temporal” would benefit from a brief discussion of whether these past studies do or don’t find effects.
4. P5: I have no specific concerns about the encouragement that participants not move their eyes, but I do wonder why this was enforced in displays that were visible for a reasonably long time (2.5s), at least enough to make several saccades. Particularly because stimuli were not scaled with distance, this choice surprised me.
5. P6: I don’t believe the manuscript mentions the cutoff for “too late” trials, which might be good to include
6. P6: How do you think restricting target locations to between 6-8 degrees of eccentricity affects your results? Doing so is similar to making targets appear more often in one quadrant of a display (only for a ring of locations around fixation), something which has been shown to speed responses to targets within versus outside the ring (Druker & Anderson, 2010, *Frontiers in Human Neuro.*). This isn’t a confound – your results still report standard CC despite this change from many CC studies – but I wonder if your results might change without that restriction in target locations (e.g., participants may be consciously guiding attention to the range of locations that could contain targets, making other implicit effects of attention less effective than if participants weren’t engaging goal-driven attention; cf. Jiang, Swallow, & Rosenbaum, 2013, *Guidance of spatial attention by incidental learning and endogenous cueing, JEP:HPP*).
7. P7: It took me a long time to learn which condition names were associated with which conditions, so you might consider renaming them to something more intuitive (e.g., old-random, old-structured, target-structured, and new, or similar).
8. P8: Your log-transforming of all RT data together, rather than separately within each condition (and perhaps within each subject and condition), may be inappropriate. Could you explain your reasoning behind log-transforming all RT data independent of condition given the possibility that these data can be viewed as coming from different distributions assuming you do have an effect of condition?
9. P9: Fig 2a says the reported RT data is smoothed but not how this smoothing is performed. It would be good to state this.
10. P9: To be consistent, it would be good to report effect sizes for F tests as well as t-tests throughout.
11. P11: Related to my point about the discussion at the end of your introduction, I don’t think your results indicate that “there is no effect of temporal predictive context in our experiment,” but that “there is no effect of adding temporal predictive context to spatial predictive context in our experiment.”
12. Do you think your results for the Targ-so condition would differ if the absolute serial order of target locations remained constant within each block (e.g., an experiment with the same sequence of target locations throughout an entire block)? I wonder if the need to identify when this sequence begins makes it very difficult to pick up on and/or guide attention based on this predictability.
13. P15: I recommend incorporating a discussion of Vadillo, Konstantinidis, and Shanks (2016, *PBR*) in your discussion of statistical power for explicit recognition tests, as it deals specifically with this question for contextual cueing tasks. This is only a suggestion.

===PREPARING YOUR MANUSCRIPT===

===PREPARING YOUR REVISION IN SCHOLARONE===

- If you are providing image files for potential cover images, please upload these at this step, and inform the editorial office you have done so. You must hold the copyright to any image provided.
- A copy of your point-by-point response to referees and Editors. This will expedite the preparation of your proof.

- Ensure that your data access statement meets the requirements at <https://royalsociety.org/journals/authors/author-guidelines/#data>. You should ensure that you cite the dataset in your reference list. If you have deposited data etc in the Dryad repository, please include both the 'For publication' link and 'For review' link at this stage.
- If you are requesting an article processing charge waiver, you must select the relevant waiver option (if requesting a discretionary waiver, the form should have been uploaded at Step 3 'File upload' above).
- If you have uploaded ESM files, please ensure you follow the guidance at <https://royalsociety.org/journals/authors/author-guidelines/#supplementary-material> to include a suitable title and informative caption. An example of appropriate titling and captioning may be found at https://figshare.com/articles/Table_S2_from_Is_there_a_trade-off_between_peak_performance_and_performance_breadth_across_temperatures_for_aerobic_scope_in_teleost_fishes_/3843624.

Author's Response to Decision Letter for (RSOS-201565.R0)

See Appendix A.

RSOS-201565.R1 (Revision)

Review form: Reviewer 2

Is the manuscript scientifically sound in its present form?

Yes

Are the interpretations and conclusions justified by the results?

Yes

Is the language acceptable?

Yes

Do you have any ethical concerns with this paper?

No

Have you any concerns about statistical analyses in this paper?

No

Recommendation?

Accept with minor revision (please list in comments)

Comments to the Author(s)

The authors have done an outstanding job in revising this paper and I think it would make a good contribution to Royal Society Open Science. I have only one minor comment.

p41: In line with revisions you've already made elsewhere (e.g., p36), I think the claim that "temporal predictive context is not exploited when locating a target in a complex scene using a typical visual search task" is not supported by your design; instead, it needs to be clarified that the Old-random vs Old-structured comparison shows the lack of this exploitation when spatial predictive context is also present.

Decision letter (RSOS-201565.R1)

Dear Ms Bouwkamp

On behalf of the Editors, we are pleased to inform you that your Manuscript RSOS-201565.R1 "No exploitation of temporal sequence context during visual search" has been accepted for publication in Royal Society Open Science subject to minor revision in accordance with the referees' reports. Please find the referees' comments along with any feedback from the Editors below my signature.

Please submit your revised manuscript and required files (see below) no later than 7 days from today's (ie 12-Jan-2021) date. Note: the ScholarOne system will 'lock' if submission of the revision is attempted 7 or more days after the deadline. If you do not think you will be able to meet this deadline please contact the editorial office immediately.

Kind regards,
Royal Society Open Science Editorial Office
Royal Society Open Science

on behalf of Dr Isabelle Mareschal (Associate Editor) and Essi Viding (Subject Editor)
openscience@royalsociety.org

Reviewer comments to Author:

Reviewer: 2

Comments to the Author(s)

The authors have done an outstanding job in revising this paper and I think it would make a good contribution to Royal Society Open Science. I have only one minor comment.

p41: In line with revisions you've already made elsewhere (e.g., p36), I think the claim that "temporal predictive context is not exploited when locating a target in a complex scene using a typical visual search task" is not supported by your design; instead, it needs to be clarified that the Old-random vs Old-structured comparison shows the lack of this exploitation when spatial predictive context is also present.

===PREPARING YOUR MANUSCRIPT===

===PREPARING YOUR REVISION IN SCHOLARONE===

To revise your manuscript, log into <https://mc.manuscriptcentral.com/rsos> and enter your Author Centre - this may be accessed by clicking on "Author" in the dark toolbar at the top of the

page (just below the journal name). You will find your manuscript listed under "Manuscripts with Decisions". Under "Actions", click on "Create a Revision".

<https://royalsociety.org/journals/authors/author-guidelines/#supplementary-material> to include a suitable title and informative caption. An example of appropriate titling and captioning may be found at https://figshare.com/articles/Table_S2_from_Is_there_a_trade-off_between_peak_performance_and_performance_breadth_across_temperatures_for_aerobic_sc_ope_in_teleost_fishes_/3843624.

Author's Response to Decision Letter for (RSOS-201565.R1)

See Appendix B.

Decision letter (RSOS-201565.R2)

Dear Ms Bouwkamp,

It is a pleasure to accept your manuscript entitled "No exploitation of temporal sequence context during visual search" in its current form for publication in Royal Society Open Science.

Before we proceed to the publication stage, we ask that you please now ensure that your Donders dataset is now made public. Please reply to this email confirming this once your data are made public.

on behalf of Dr Isabelle Mareschal (Associate Editor) and Essi Viding (Subject Editor)
openscience@royalsociety.org

Appendix A

To the reviewers,

We would like to sincerely thank the reviewers for their positive evaluation of and constructive feedback on our manuscript. We believe this greatly improved the manuscript. Below we answer the questions raised and specify how we implemented suggested changes. Textual changes in the manuscript have been copy-pasted and highlighted in yellow. Page numbers refer to the revised manuscript.

Reviewer: 1

Comments to the Author(s)

This study aims at investigating the effect of temporal regularities of events on human behavior. Using a modified version of the classic spatial contextual cueing paradigm, they show that temporal context – manipulated as the predictive order of search arrays - does not modulate response times. The experiment design and data analysis are carefully conducted, and the findings are unquestionably relevant to the field. I have only a data analysis suggestion and a more general comment.

A possible limitation of their analysis is that it's not possible to know when - within each sub-block - the temporal context would kick in. The authors somewhat acknowledge that, by excluding the first trial of each sub-block. However, I believe that there are better ways to deal with this potential problem. The simplest one would be to analyze RT performance separately for each display repetition (1 to 6). The other, perhaps more sensitive approach, would be to perform a regression analysis using the order as a predictor.

Thank you for this suggestion. It is indeed unknown when temporal context would impact behavior. What is certain, however, is that the first display of the sequence cannot be predicted based on the preceding display. That was the rationale for excluding this display from the analyses. It could indeed be that the impact of temporal context on behavior is only visible later in the sequence. To investigate this possibility, we ran an analysis of variance on the Old-so – Old-ro effect per display. More specifically, we preprocessed the data as before, and focused on correct responses in the second half of the experiment to be optimally sensitive to a possible effect of learned predictive context. Per subject, we then calculated the average reaction time per display in the Old-so condition and subtracted the average reaction time of the Old-ro condition across displays. We then ran a One Way Repeated Measures ANOVA with the So-Ro effect as dependent variable and display number as a within subject factor. If the impact of temporal context develops over time, we would expect the So-Ro effect to depend on display number. This is, however not what we find ($F_{5,185} = 0.30$, $p = 0.91$).

Below we plot the RO-SO effect in reaction times for each of the displays.

We hope this extra analysis satisfactorily addresses this point brought up by the reviewer. We added this analysis alongside a similar analysis upon the request of reviewer 2 to the manuscript (p10) :

“It might be that temporal predictive context requires time to be recognized, and only impacts reaction time later in the sequence. To examine this possibility, we analyzed the effect of temporal context in structured conditions per display (Old-structured per display versus Old-random overall; Target-structured per display versus New overall). Neither of these effects depended on display number (Old-so $F_{5,185} = 0.30$, $p = 0.91$, and Targ-so $F_{5,185} = 1.07$, $p = 0.38$).”

As a more general comment, the authors should be a bit more careful when they generalize their findings. Temporal context is an extremely broad concept, and I don't believe that a manipulation of the sequence order of events is enough to state that temporal context is not exploited, or extracted, during visual search. One of the original papers in this area (Olson and Chun, 2001), already noted the intricacies of the term: “Temporal structure can be divided into two time-based variables: (a) a sequential or ordinal variable and (b) a duration or rhythm-based variable.” (pg. 1301). Therefore, I would strongly suggest to the authors to be more specific about the type of temporal prediction they investigated (sequence order) - including in the title and abstract.

We agree with the reviewer that we should have been more specific in our definition of temporal context. Our experimental manipulation of temporal context is based on sequence order, and we have now clarified this in the manuscript. To this end, we adjusted the abstract, introduction and discussion section. The new title of the manuscript is

“No exploitation of temporal sequence context during visual search”. We also added the following to the discussion (p16):

“Our results have implications for how we think about the role of predictive context in visual perception. Though both spatial and temporal context are of importance for visual perception (38), our findings imply that extracting and using context might not be as automatic as previously assumed. In our experiment we find that spatial predictive context is favored. This might be due to a modality-specific bias, leading to a dominance of spatial predictive context in visual perception (39–43). However, sensitivity to one type of context over the other might also be task-dependent. Additionally, there are other types of temporal context, for instance duration- or rhythm-based (29). These temporal predictive contexts create expectations within the temporal domain itself: when the target will occur, instead of where. These types of temporal context might be exploited more efficiently in addition to spatial predictive context.”

Reviewer: 2

Comments to the Author(s)

Summary: This paper reports an experiment on the relationship between spatial and temporal predictive information in visual search. Participants did a T-among-L search task that had 4 conditions (across sub-blocks): (1) random order and configuration of all search items; (2) random order of 6 trials with repeated target and distractor locations; (3) repeated order of 6 trials with repeated distractor and target locations; and (4) repeated order of 6 trials with repeated target locations (but not distractor locations).

This paper investigates an interesting question and is well-designed. I like this study, and I think the writing is strong: the introduction in particular is highly focused, which makes it easy to grasp the point of the study (with some minor exceptions I mention below), and the general discussion covers the relevant literature well. My concerns with this version of the study deal primarily with the analyses and some of the theoretical interpretation, both of which I think could be greatly improved with a revision to this submission.

Major Points:

1. I think your analysis of individual differences is inappropriate. By combining Old-ro and Old-so trials and comparing to New, then analyzing the difference within participants who show no Old vs. New difference, you are ensuring that on average participants in the “no contextual cueing” group show no Old-ro versus Old-so difference. The only way for them to show no Old vs New difference but a difference between Old-ro and Old-so is if one of these two Old conditions is worse than the new condition, something there is (to my knowledge) no theoretical reason to expect—

this would only be due to noise. Therefore, I don't think this test can appropriately assess whether participants who don't show standard contextual cueing will show an advantage of repeating temporal context. Rather, you might use correlational analyses similar to those you show in Fig 3. For instance, you could compute the standard CC effect (Old-ro - New) and plot this against the Old-so effect (e.g., Old-ro - Old-so), which as far as I can tell avoids the concern I mention above. This is only one suggestion: there may be other, more appropriate ways to conduct these individual difference analyses. However, I do feel that a different approach to the one currently used is necessary.

We are thankful for the reviewer for pointing this out. We agree that within the No-contextual cueing group the contrast between Old-ro and Old-so is biased towards finding no effect. In the initial stages of data analysis we ran exactly the correlation analysis suggested by the reviewer, but soon realized that such an analysis will always produce spurious (false) positive results. This is caused by both difference scores (Old-ro - New) and (Old-ro - Old-so) being partly based on data from the same condition (Old-ro). Below a simple matlab simulation and plot based on pure noise demonstrating this issue:

```
%generate three variables with 100 random numbers coming from a normal distribution
```

```
Old_so = randn(100, 1);  
Old_ro = randn(100, 1);  
New = randn(100, 1);
```

```
%generate effects (difference scores)
```

```
CC = Old_ro - New;  
RoSo = Old_ro - Old_so;
```

```
%calculate correlation
```

```
[r, p] = corr(CC, RoSo);  
%r = 0.550  
%p = 3.0283e-09
```

```
%scatterplot
```

```
figure;  
scatter(CC, RoSo, 'filled');  
title('scatterplot of effects'); xlabel('CC effect'); ylabel('RoSo effect');
```

Thus showing that the suggested analysis is unfortunately not feasible.

Importantly, the individual difference analysis was primarily focused on assessing a potential effect of Old-so vs Old-ro within the group that **did** show contextual cueing. As stated in the discussion, our reasoning was as follows (p14):

“Our main finding is that sequence-based temporal predictive context was not exploited during our visual search task. It is important to acknowledge that visual search is an inherently spatial task. Even in the temporal predictive context conditions, spatial location(s) predicted spatial location(s) on the next trial. From this, it follows that in order to even detect temporal context, spatial context needs to be processed first. Therefore, perhaps only participants who are sensitive to spatial predictive context, are also sensitive to temporal predictive context.”

Testing for an Old-so vs Old-ro effect within the contextual cueing group does not suffer from the (conservative) bias identified by the reviewer when testing for this effect within the **no** contextual cueing group. Since the contextual cueing group was the group of interest to begin with, we have now chosen to focus on it exclusively, and have removed the analyses on the no contextual cueing group altogether. (These were presented originally mainly for completeness' sake, but given these issues we agree it is better to omit them.) In the manuscript the paragraph now reads as follows (p14):

“It is possible that the tendency to benefit from both temporal and spatial predictive context is governed by individual differences. Perhaps only people who show sensitivity to spatial predictive context are likely to be sensitive to temporal predictive context. To explore this possibility, in a follow-up analysis, we focused specifically on those participants sensitive to spatial predictive context (i.e., those with a contextual cueing effect in general, as defined by a RT benefit for Old displays (Old-random/Old-structured combined) compared to New). Within this group, we tested if there was a difference between the Old conditions (Old-random versus Old-structured).”

2. Throughout the paper, I think you can be more precise regarding what your findings say about temporal predictability. The introduction seems to inconsistently discuss the effects of temporal predictability on visual search in general and the interaction between spatial and temporal predictability. I think revision of the last couple paragraphs of the introduction to lay out the fact that your experiment fully crosses manipulations of temporal predictability of targets (albeit target locations) with spatial predictability of entire search arrays. My impression is that you are addressing two interesting questions: (1) will temporal predictability help (in which case Targ-so condition should be faster than New and (2) will temporal predictability help when also in the presence of spatial predictability (Old-so faster than Old-ro). As

currently written, your preview of the results makes it sound like you are only testing (2) and based on it concluding that temporal predictability is not useful in your task, while in reality you do present marginal evidence that temporal predictability is useful relative to no predictability, just that it is redundant with spatial predictability.

We appreciate this comment and the viewpoint that we are missing out on an additional interesting research question. We have, however, some reservations that keep us from making the claim that we used a fully crossed design. Furthermore, we are reluctant to claim that our data provide evidence for an effect of temporal predictability without spatial predictability (Targ-so). We initially included the Targ-so condition as a control condition, as discussed on P15:

“We included this condition to be able to address the question to what extent a possible effect of temporal predictive context elicited by the entire scene (Old-so) might be due solely to the repetition of the target, independent of the rest of the spatial configuration.”

As such, we were surprised to find a marginal effect of Targ-so vs New. While considering the most likely interpretation of this effect, we realized that we were restricting possible target locations in all experimental conditions, except New. I.e., on a given trial, the target location would necessarily be drawn from a set of 6 possibilities in Old-ro, Old-so, and Targ-so, whereas it would be drawn from many more possibilities in New. This makes the comparison between Targ-so and New a slightly biased one as people were more likely to encounter a less familiar target location in the New condition. If there is a true effect of Targ-so vs New (for which we have doubts in the first place—see below), we believe the most likely interpretation is an effect of location familiarity (i.e. overall target location probability), rather than of sequence-based temporal predictability.

Relatedly, we therefore cannot claim to have a fully crossed design: replacing the New condition with a Targ-ro condition (with the same restricted set of possible target locations, but with no structured order) would have come closer to such a crossed design. However, in order to facilitate the interpretability of our findings within the broader literature on contextual cueing, we decided to stay relatively close to this literature, in which the typical comparison (in our terms) is between Old-ro and New.

(It should be noted that the potential interpretation we describe above, related to target location familiarity, might also be said to hold for the comparison between Old-ro and New. However, the effect here is of substantial size, while target location familiarity, if at all, maximally contributes only a small effect (as evidenced by the Targ-so vs New contrast). We therefore deem the traditional interpretation of the Old-ro vs New contextual cueing effect in terms of spatial context-based predictability most likely.)

We moreover have serious doubts as to whether there is a true effect of Targ-so vs New in the first place. The statistical test of this effect in reaction times had a p-value of 0.05, strictly speaking not crossing the threshold for ‘significance’ ($p < 0.05$). Furthermore, Bayesian analysis indicated inconclusive evidence ($BF = 1.02$). The effect in accuracy was

more clearly non-significant ($p = 0.17$), with anecdotal evidence in favour of the null hypothesis ($BF = 0.43$). In contrast, for both dependent variables, the effect of spatial predictive context was highly significant, with very strong evidence against the null hypothesis.

Taken together, we are confident about the reported effect of spatial predictive context, but are not convinced (a) that the marginal effect in the Targ-so condition reflects a true effect, rather than a chance finding; and even if it does, (b) that it reflects temporal-order based predictability. This is why the main research question is (p3):

“Can observers learn from both spatially predictive and temporally predictive context in visual search?”

And our main conclusion is (abstract):

“We argue that spatial predictive context during visual search is more readily learned and subsequently exploited than temporal predictive context, potentially rendering the latter redundant.”

It is not our intention to claim with confidence that there is no effect of temporal predictive context generated by target repetitions in a sequenced order per se, given the uncertainty outlined above, and the fact that our experiment was not specifically designed to test this. This is why we adjusted the discussion about this finding as follows (p15):

“We found a marginal effect when only target locations were presented in a structured order (Targ-so). We do not feel confident interpreting this result, as it was only present in reaction time and not in accuracy, strictly speaking did not cross the threshold for significance ($p < 0.05$), and Bayesian analysis indicated inconclusive evidence. We included this condition to be able to address the question to what extent a possible effect of temporal predictive context (if found) elicited by the entire scene (Old-so) might be due solely to the repetition of the target, independent of the rest of the spatial configuration. By design, the Targ-so condition had a limited set of target locations compared to the completely novel displays, which in turn would elicit a small amount of spatial predictive context. If the marginal benefit we observed indeed reflects a true effect, we believe this might explain it.”

Minor points:

3. P4: the paragraph beginning “to investigate temporal” would benefit from a brief discussion of whether these past studies do or don’t find effects.

We have added a brief summary of what the cited papers found with regard to temporal predictive order based on sequence at the suggested location (p3):

“To investigate temporal predictive context based on sequences, single items are generally shown in a specific order, i.e. as a pair, triplets, or longer sequences. Due to this order, one item becomes predictive of the following item(s). These items can be syllables (12), shapes (6), or objects (13–15). In these studies, ranging from infants to primates, temporal predictive context typically leads to improved visual processing. Temporally predicted targets are more familiar, and detected or categorized faster (6, 15) than unpredicted targets. In the brain, temporally expected images are accompanied by neural signatures of improved visual processing such as a higher dynamic range (13–14) or a suppressed BOLD response (15) compared to unexpected images. Similarly to the spatial regularities in contextual cueing, this temporal regularity typically goes unnoticed by participants.”

4. P5: I have no specific concerns about the encouragement that participants not move their eyes, but I do wonder why this was enforced in displays that were visible for a reasonably long time (2.5s), at least enough to make several saccades. Particularly because stimuli were not scaled with distance, this choice surprised me.

The reason why we enforced fixation was because this study was a prelude to a potential study using Magnetoencephalography (MEG) and we wanted the study design to be as similar as possible to this potential follow-up. Eye movements are a serious confound in MEG measurements so they are typically prevented by enforcing fixation. We knew from previous work, both by others and in our lab, that the contextual cueing remains intact under fixation.

van Asselen M, Castelo-Branco M. The role of peripheral vision in implicit contextual cuing. *Perception & Psychophysics*. 2009 Jan 1;71(1):76-81. <https://doi.org/10.3758/APP.71.1.76>

Spaak E, de Lange FP. Hippocampal and prefrontal theta-band mechanisms underpin implicit spatial context learning. *J Neurosci*. 2020 Jan 2;1660–19. <https://doi.org/10.1523/JNEUROSCI.1660-19.2019>

5. P6: *I don't believe the manuscript mentions the cutoff for "too late" trials, which might be good to include*

Thank you for pointing this out. Participants were too late when the visual search display was no longer on the screen, which was maximally 2.5 sec. We adjusted the manuscript as follows, hopefully making this more clear (P4):

"Each trial started with a 1 s fixation period, and search displays were presented until the response button press or up to 2.5 s, after which responses were too late. After the button press, participants were informed whether their response was on time and correct or not by the outer part of the fixation dot turning green (correct), red (incorrect) or blue (too late) for 0.5 s (Figure 1A)."

6. P6: *How do you think restricting target locations to between 6-8 degrees of eccentricity affects your results? Doing so is similar to making targets appear more often in one quadrant of a display (only for a ring of locations around fixation), something which has been shown to speed responses to targets within versus outside the ring (Druker & Anderson, 2010, *Frontiers in Human Neuro.*). This isn't a confound—your results still report standard CC despite this change from many CC studies—but I wonder if your results might change without that restriction in target locations (e.g., participants may be consciously guiding attention to the range of locations that could contain targets, making other implicit effects of attention less effective than if participants weren't engaging goal-driven attention; cf. Jiang, Swallow, & Rosenbaum, 2013, *Guidance of spatial attention by incidental learning and endogenous cueing, JEP:HPP*).*

This is an interesting point. The reason for the restriction was to prevent both the target being too close to fixation and thus too easily detected, and it being too peripheral and thus too hard to detect. Equating search difficulty among the scenes to some degree (and thus minimizing variability) was important considering the fact that we had only 6 displays/target locations per condition. As the reviewer correctly points out, this restriction is not a confound as we applied it to all conditions equally, including New.

The question that remains is then whether having a higher probability region, potentially leading to what is known as 'spatial probability cueing', would diminish the effect of spatial and/or temporal predictive context. Such rivalry of effects was indeed found by Jiang, Swallow & Rosenbaum (2013), where an endogenous cue eliminated spatial attentional guidance by probability cueing.

Spatial probability cueing and contextual cueing are distinct but very similar effects, and it has been argued they both create 'search habits' (Jiang, 2018). Since neither fall under the notion of goal driven attention (unlike endogenous cueing) we feel it is unlikely these two effects are at arms with each other. It could be that temporal context only impacts search behavior when it becomes explicit and can drive attention in a goal-driven way. This would be an interesting follow up question. We added this to the discussion section (p16)

“...Signs of sequence learning were visible after 96 repetitions (35). It is possible that this many repetitions of the spatial context in our study would have allowed exploitation of temporal predictive context. We question, however, whether learning will then remain implicit, introducing another source of variance. Nevertheless, it is a potentially interesting question: it could be that when temporal predictive context becomes explicit it will impact reaction times via goal-directed attention (36). On a very speculative note, temporal predictive context might even become an endogenous cue and block the more habitual effect of spatial predictive context (37).”

36. Jiang YV, Swallow KM, Rosenbaum GM. Guidance of spatial attention by incidental learning and endogenous cuing. *Journal of Experimental Psychology: Human Perception and Performance*. 2013 Feb;39(1):285. <https://doi.org/10.1037/a0028022>

37. Jiang YV. Habitual versus goal-driven attention. *Cortex*. 2018 May 1;102:107-20. <https://doi.org/10.1016/j.cortex.2017.06.018>

7. P7: It took me a long time to learn which condition names were associated with which conditions, so you might consider renaming them to something more intuitive (e.g., old-random, old-structured, target-structured, and new, or similar).

We changed condition labels throughout the text and figures to Old-random, Old-structured, Target-structured and New.

8. P8: Your log-transforming of all RT data together, rather than separately within each condition (and perhaps within each subject and condition), may be inappropriate. Could you explain your reasoning behind log-transforming all RT data independent of condition given the possibility that these data can be viewed as coming from different distributions assuming you do have an effect of condition?

We log-transformed the RT data because reaction times are typically non-normally distributed and heavily skewed, and this transformation substantially improves normality. This transformation of the reaction times was trial-based: we computed log₁₀ of the response time per trial. This adjustment involves no normalization relative to any particular distribution. (In other words, log-transformation per condition, or independently of conditions, are equivalent.) We adjusted our description of this procedure in the manuscript, which hopefully prevents the confusion (p7):

“Since RT distributions are typically heavily skewed, the reaction times per trial were log-transformed by taking the log₁₀ value prior to any statistical analysis to improve normality.”

9. P9: Fig 2a says the reported RT data is smoothed but not how this smoothing is performed. It would be good to state this.

Thank you for pointing this out. We now explain the smoothing procedure not only in data analysis part of the main text (p7), but also in the figure caption(p9):

Figure 2. Results main search task (A) Reaction time (smoothed across neighbours, by taking the mean of block N, N-1 and N+1) plotted over the timecourse of the experiment (shading indicates within-participant corrected standard error of the mean) (B) Reaction time and (C) Accuracy within the second half of the experiment (block 19-36). Colored dots are individual participants, the black dot reflects the mean, and the black bar indicates ± 1 standard deviation.

10. P9: To be consistent, it would be good to report effect sizes for F tests as well as t-tests throughout.

We now report effect sizes for all significant effects and added the following to the manuscript (p8):

“We report raw RTs in the Results section, but all statistical tests were done on log-transformed and QF-corrected quantities. If the assumption of sphericity was violated, as indicated by a significant outcome of Mauchly’s test, we report the corrected p value and degrees of freedom. All Greenhouse-Geisser ϵ values were above 0.75, we therefore reported the more liberal Huyn-Feldt corrected values (20). Effect sizes were calculated for all significant effects using the effsize package for R. For F-tests, this is the Generalized Eta-Squared measure of effect size (21). For T-tests the value of Cohen’s d is computed using the approach of Gibbons et al. (22) for paired samples, including a suggested correction of Borenstein (23).”

21. Bakeman R. Recommended effect size statistics for repeated measures designs. Behavior research methods. 2005 Aug 1;37(3):379-84.

11. P11: Related to my point about the discussion at the end of your introduction, I don’t think your results indicate that “there is no effect of temporal predictive context in our experiment,” but that “there is no effect of adding temporal predictive context to spatial predictive context in our experiment.”

In line with our earlier response, we agree with this. We wish to restrict our conclusion to the effect of Old-so. To avoid redundancy we combined the last two sentences of this paragraph as follows (p10);

“Importantly, testing the key hypothesis of our experiment, we find no benefit of presenting scenes in a predictable order. Participants were on average equally fast in finding the target in both type of Old displays (Old-ro: 1.16 ± 0.44 ms, Old-so: 1.15 ± 0.44 ms, $t_{37} = 0.7$, $p = 0.50$). We find similar results for accuracy (Old-ro: $91.34 \pm 8.51\%$, Old-so: $91.83 \pm 9.01\%$, $t_{37} = -0.6$, $p=0.56$). The Bayesian analysis of these results ($BF_{10} = 0.22$ for reaction time and $BF_{10} = 0.21$ for accuracy) indicate that the data are 4 – 5 times more likely under the null than under the alternative hypothesis. From this we can conclude that there is no effect of adding temporal predictive context to spatial predictive context in our experiment.”

12. Do you think your results for the Targ-so condition would differ if the absolute serial order of target locations remained constant within each block (e.g., an experiment with the same sequence of target locations throughout an entire block)? I wonder if the need to identify when this sequence begins makes it very difficult to pick up on and/or guide attention based on this predictability.

This is a very interesting thought. Such a design might indeed make participants more sensitive to temporal predictive context. However, we question whether this would remain implicit. It would be an interesting research question for a follow-up experiment with fewer conditions, for instance only contrasting Targ-so and Targ-ro.

Regarding our experiment: if the need to identify when the sequence begins prevents temporal predictability from facilitating search, we would expect the effect to maybe become visible as you are further in the sequence. This is related to the idea of temporal predictive context to ‘kick in’ later, raised by reviewer 1 (point 1). Therefore, we performed a similar analysis. We ran a regression analysis on the Targ-so–New effect per display. More specifically, we preprocessed the data as before, and focused on correct responses in the second half of the experiment to be optimally sensitive to a possible effect of learned predictive context. Per subject, we then calculated the average reaction time per display in the Targ-so condition and subtracted the average reaction time of the New condition across displays. We then ran a One Way Repeated Measures ANOVA with the Targ-so - New effect as dependent variable and display number as a within subject factor.

If it takes time to recognize the sequence and subsequently for temporal predictive context to have an effect, we would expect an this effect to depend on display number. This was not the case ($F_{5,185} = 1.07$, $p = 0.38$)

Below we plot the effect per display number.

This does not exclude the proposed possibility that a different, more strictly repetitive, design would reveal an effect of course, but at least we do not find evidence for such a ‘kicking in’ effect in our data.

We added this analysis alongside a similar analysis upon the request of reviewer 1 to the manuscript (p10):

“It might be that temporal predictive context requires time to be recognized, and only impacts reaction time later in the sequence. To examine this possibility, we analyzed the effect of temporal context in structured conditions per display (Old-structured per display versus Old-random overall; Target-structured per display versus New overall). Neither of these effects depended on display number (Old-so $F_{5,185} = 0.30$, $p = 0.91$, and Targ-so $F_{5,185} = 1.07$, $p = 0.38$).”

13. P15: I recommend incorporating a discussion of Vadillo, Konstantinidis, and Shanks (2016, PBR) in your discussion of statistical power for explicit recognition tests, as it deals specifically with this question for contextual cueing tasks. This is only a suggestion.

We agree with this recommendation, and discuss this particular paper in the Discussion (P14):

“Therefore, we might have been under-powered to detect above chance performance with this relatively small sample of displays (25).”

25. Vadillo MA, Konstantinidis E, Shanks DR. Underpowered samples, false negatives, and unconscious learning. Psychon Bull Rev. 2016 Feb;23(1):87–102.

Appendix B

To the reviewer,

We would like to sincerely thank reviewer 2 for their positive evaluation of our revision of the manuscript. Below we address the minor comment. Textual changes in the manuscript have been copy-pasted and highlighted in yellow. Page numbers refer to the revised manuscript.

Reviewer comments to Author: Reviewer: 2

Comments to the Author(s)

The authors have done an outstanding job in revising this paper and I think it would make a good contribution to Royal Society Open Science. I have only one minor comment.

p41: In line with revisions you've already made elsewhere (e.g., p36), I think the claim that "temporal predictive context is not exploited when locating a target in a complex scene using a typical visual search task" is not supported by your design; instead, it needs to be clarified that the Old-random vs Old-structured comparison shows the lack of this exploitation when spatial predictive context is also present.

In line with this comment we revised the manuscript as follows (P41/pnr.14);

"We found a clear and strong benefit for old displays compared to novel displays, confirming the strong influence of spatial predictive context during visual search. However, we did not observe an effect of repeating old scenes in a consistent order compared to presenting them in a random order. This was also true when we focused on only those participants who were sensitive to spatial predictive context. We thus found evidence that, using a typical visual search task, when locating a target in a complex scene, temporal predictive context is not exploited when spatial predictive context is present. We discuss these findings below."